# A Novel Resource Allocation Scheme in NOMA-Based Cellular Network with D2D Communications

**Jingpu Wang**, **Xin Song \* and Yatao Ma**

Engineering Optimization and Smart Antenna Institute, Northeastern University, Qinhuangdao 066004, Hebei, China; 1610533@stu.neu.edu.cn (J.W.); 1871644@stu.neu.edu.cn (Y.M.)
* Correspondence: sxin78916@neuq.edu.cn

**Abstract:** Non-orthogonal multiple access (NOMA) has become a promising technology for 5G. With the support of effective resource allocation algorithms, it can improve the spectrum resource utilization and system throughput. In this article, a new resource allocation algorithm in the NOMA-enhanced cellular network with device-to-device (D2D) communications is proposed, in which we use two new searching methods and an optimal link selection scheme to maximize the system throughput and limit the interferences of the NOMA-based cellular network. In the proposed joint user scheduling, tree-based search power allocation and link selection algorithm, we simplify the solving process of previous methods and set up the optimization function, which does not need to be derivable. With successive interference cancellation (SIC) technology, we give conditions for the D2D devices accessing into the network. We also propose a suboptimal scheme to schedule cellular users and D2D devices into multiple subchannels, which reduces the complexity of the exhaustive search method. Through consistent tree-based searching for the power allocation coefficients, we can get the maximum arithmetic average of the system sum rate. Meanwhile, for the existence of the part of interferences from larger power users which can be canceled by the SIC in NOMA systems, the search options are decreased for increasing the search rate of the power allocation algorithm. Moreover, we propose a distance-aware link selection scheme to guarantee the quality of communications. In summary, the proposed algorithm can improve the system throughput, has a low complexity cost and potentially increases spectral utilization. Numerical results demonstrate that the proposed algorithm achieves a higher data transmission rate than some of the traditional methods and we also investigate the convergence and the computational complexity cost of the joint algorithm.

**Keywords:** non-orthogonal multiple access (NOMA); device-to-device (D2D); successive interference cancellation (SIC); user scheduling; tree-based search power allocation algorithm

## 1. Introduction

Non-orthogonal multiple access (NOMA) has recently attracted attention from academic communities as a novel energy and spectrum efficient technology due to a higher network capacity compared with orthogonal multiple access (OMA) in the fifth generation (5G) environment [1]. NOMA networks are expected to deliver real-time contents such as monitoring and multimedia streams, and non-real-time contents such as web browsing, images, messaging, and file transfers for users [2,3]. Multi-carrier NOMA (MC-NOMA) along with sparse code multiple access (SCMA) and pattern division multiple access (PDMA) have been comprehensively investigated on the basic principles and enabling schemes in [4]. There are also many articles concerning NOMA. Some works [5,6] pay close attention to the quality of service (QoS) parameters such as SINR of channels and capacity

to try to promote the spectral efficiency performance. In [7], with the concept of MIMO-NOMA, some key technical problems in the system are summarized. Moreover, an important issue of successive interference cancellation (SIC) is put forward and the future research directions are presented in this area. In some essays [8,9], to improve QoS, a NOMA radio network with simultaneous wireless information and power transfer is studied under a non-linear energy harvesting model. In [10], a method for predefining a minimum transmission rate for each user to guarantee QoS is focused on. In [11,12], a new secrecy transmission paradigm and an advanced resource allocation algorithm for uplink and downlink NOMA systems are proposed respectively. The energy efficiency (EE) is studied in a NOMA enabled heterogeneous cloud radio access network (H-CRAN) in [13] in which key technologies in 5G network are discussed to be properly implemented that can be applied in NOMA H-CRANs to improve EE. Most of the algorithms mentioned above show that NOMA technology is capable of satisfying the requirements of the 5G wireless communication standard from different aspects, especially in promoting EE and spectrum efficiency (SE) and supporting more network links.

In addition to NOMA, device-to-device (D2D) communication has been an essential way to alleviate the upcoming traffic pressure on the near future networks. Owing to the rapid development of radio resource management algorithms and new peer discovery methods, D2D communication has made a significant contribution to increasing SE and EE of a 3GPP (3rd Generation Partnership Project) Long Term Evolution system by sharing spectrum resources with cellular users [14]. By using D2D technology, cellular network users can directly communicate with each other, and thus it offloads data traffic of the base station (BS) in a more and more dense cellular network. Recently, many works have suggested combing D2D with other technologies in different environments [15,16]. From the following literatures, we can also draw a conclusion that D2D is closely integrated with other communication technologies [17–22]. In [17], a novel approach for discovering indoor peers is proposed which is proved to be highly energy efficient and interference limited. In addition, many papers are concerned with applying D2D to full-duplex relay systems [18,19]. In [20], game theory is used in joint power allocation and channel selection under D2D communication scenarios. A priority based joint power control and resource allocation algorithm is proposed for enhancing EE through SIC technology under D2D-aided heterogeneous networks [21]. In [22], both beamforming and interference cancellation (IC) strategies were investigated to improve performance optimization of the D2D enhanced cellular network assisted by a two-way decode-and-forward relay node.

The promising applications of NOMA technology in D2D communications have been put forward to further improve the potential benefits of EE and SE from the algorithms mentioned above and many models with excellent technologies have been presented [23–29]. In [23,24] models of NOMA-based D2D communications for cooperative relaying systems are proposed. In [25,26] the systems are also combined with energy harvesting. Unlike the traditional concept of "D2D pair", the concept of "D2D group" in which several D2D receivers are capable of receiving information from one D2D transmitter is presented in [27]. In [28], the resource allocation problem of a NOMA-based cellular network is modeled as a Lagrangian function with KKT conditions, in which there are only two D2D power parameters. Different from the matching method of channel assignment for D2D users in [27], the optimization problem is solved by the sub-gradient method [29]. Through the above analysis, we note that NOMA can provide a fair transmission condition with Pareto optimality in power allocation from the game theory and D2D communications are effective means to improve the network capacity through increasing the number of accessed user devices.

However, the studies mentioned above rely on perfectly transforming the utility functions into convex programming problems. Moreover, they always need a large number of iterations and derivations to get the results. This directly increases the computational complexity cost, and therefore require powerful computing equipment which needs to support MATLAB and other computing software to solve the problems. For example, a group of people go camping in a remote place and they need to temporarily build a high QoS communication mode, but their devices are unable to provide enough computing power. To account for the above problem, we need to adopt a kind of algorithm to

solve the problem when it is non-differentiable or non-convex. In [30] a low computational complexity power assignment method is presented for a NOMA system which is called the tree-based search algorithm. Some research has made a further improvement on reducing the computation load and thus decreasing the computational complexity cost [31]. However, both of the papers seem to neglect the aspect of maximizing the sum data rate of the network. The object of our proposed algorithm is to keep the balance of maximizing the user throughput and lowering the computational complexity cost. Compared with the exhaustive search algorithm (ESA) [32], our method largely reduces the computational complexity without significant throughput decline. Besides that, we extend the one subchannel power allocation in [31] to a general case, in which the D2D users can be assigned to multiple subchannels.

Recall that, although D2D can unprecedentedly increase the spectrum efficiency, it divides part of the energy from the cellular network [27,33]. As D2D links reuse the same spectrum allocated to the cellular users, they may impose more interference on the network [34–37]. To mitigate the two problems, we propose a user scheduling scheme and a D2D link selection scheme. To maximize the total power of the whole cell users, we use SIC technology to calculate a threshold for the transmission rate of D2D. To the best of our knowledge, the existing works cannot use joint user scheduling, tree-based search power allocation and link selecting algorithm in NOMA and D2D enhanced multiple subchannels cellular communication systems. Considering all the problems mentioned above, the proposed algorithm first improves the user data rates, then allocates power to all the users in the network for a further throughput improvement and finally facilitates a high-quality D2D link.

In this paper, we consider a NOMA-based single-cell cellular network with D2D communications on multiple subchannels, in which a D2D device can reuse the same subchannel occupied by a cellular user to improve the spectrum utilization. Because D2D users result in interference with cellular networks, we use SIC technology to impose restrictions on the energy consumption. The main contributions of this work can be summarized as follows:

(1) The proposed algorithm can jointly solve the user scheduling, power allocation and link selection problems for the D2D underlaying cellular network with the NOMA technology, which is a candidate technology for future networks. The D2D communication is introduced to offload traffic from the base station (BS) and increase network capacity.

(2) A low computational complexity cost search algorithm has been given. Compared with the ESA, it reduces the number of searches by considering the SIC decoding order and thus improves the search rate. It is analytically proved that compared with ESA in OMA, the proposed method can reduce the computational complexity cost. Because of the way of searching for solutions without derivation, it becomes easy for the algorithm to give an optimal solution.

(3) We use the geometric mean value and the arithmetic mean value of the data rates as two objective functions of the tree-based search algorithm, respectively. The former considers the impact of the mean value when extremely high or low power signals exist, while the latter reflects the real mean value of the sum rate.

(4) The proposed joint algorithm can achieve a high data rate and achieves a more superior performance compared to other searching methods [38–41]. In addition, we can prove that the proposed algorithm converges to a stable state within limited iterations.

The rest of the paper is organized as follows. The channel model and problem formulation are introduced in Section 2. The proposed joint user scheduling, tree-based search power allocation and link selecting algorithm is elaborated in Section 3. In Section 4, the simulation results are presented, while Section 5 finally draws conclusions of the paper.

## 2. Network Model

### 2.1. Channel Model

We focus on a NOMA-based single-cell downlink scenario which requires a relatively fair way to allocate power to the devices to improve the system capacity and we also use D2D communications to further improve the SE. We consider the elastic (or non-real time) services in network data transmission, which are shiftable in time and delay, to be tolerant. Our utility function is log-based, which means that the higher the data rate that is allocated to the user by the system, the more his utility is increased [3]. In this network, we assume that BS cannot get the perfect channel state information (CSI) and serve cell users (CUs) through $M$ subchannels (SCs) which are orthogonal, i.e., $SC_m \in SC$, $SC = \{SC_1, \ldots, SC_m, \ldots, SC_M\}$. On the same SC of the network, the interference is divided into two parts. The interference received at a D2D receiver (DR) comes from the BS (the long dashed line) and the D2D interference (the short dashed line) represents the interference from a D2D transmitter (DT) to other CUs (as shown in Figure 1a).

In Figure 1b, we consider that, the CUs of $m-$th SC (or SC$m$), $n$CUs are multiplexed on the same SC and split in the power domain by adopting NOMA. We denote $N = \{1, \ldots, n\}$ as the set of CUs. We denote $L = \{1, \ldots, l\}$ as the set of D2D users (DUs). The superposition symbol transmitted by BS on SC$m$ to CU$i$ is

$$y_{i,m} = h_{i,m} \sum_{k=1}^{n} s_{k,m} \sqrt{p_{k,m}} + z_{i,m},$$ (1)

where $s_{k,m}$ is the transmitting signal for CU$k$, $p_{k,m}$ represents the transmit power for CU$k$, and $h_{i,m}$ denotes the channel gain from BS to CU$i$ on SC$m$. The receivers are assumed to have the imperfect CSI by channel feedback. Meanwhile, the noise term $z_{i,m}$ is a zero-mean complex additive white Gaussian noise (AWGN) at the BS on SC$m$ with variance $\sigma^2$.

It is assumed that the cellular users and D2D transmitters are uniformly distributed in the cell. We also assume that channels between CUs and BS are undergoing a path-loss model with slow fading caused by shadowing and fast fading caused by the multi-path propagation. The channel coefficient is constant for each channel. Thus, the channel gain from BS to CU$i$ on SC$m$ can be expressed as

$$h_{i,m} = \kappa \tau_{i,m} \varsigma_{i,m} d_{i,m}^{-\alpha}$$ (2)

where $\kappa$ denotes the constant path loss coefficient determined by system parameters, $\tau_{i,m}$ is the fast fading gain with exponential distribution, $\varsigma_{i,m}$ is the slow fading gain with log-normal distribution, $d_{i,m}$ is the distance between CU$i$ and the BS, and $\alpha$ denotes the path loss exponent.

In general, the distance between the DUs is not so far as that between cellular users and the BS. In here, we just consider fast fading for DUs. The channel gain of the $l-$th DU on SC$m$ can be expressed as

$$g_{l,m} = \kappa \tau_{l,m} d_{l,m}^{-\alpha}$$ (3)

where $\tau_{l,m}$ is the fast fading gain with exponential distribution, $d_{l,m}$ is the distance between the $l-$th D2D pair.

In practice, perfect channel state information (CSI) is not usually available. To characterize the channel condition, we apply the minimum mean-square error (MMSE) channel estimation in the channel model. The MMSE estimator employs second order statistics which involve using the channel auto covariance in order to minimize the square error. Here the channel second order statistics are assumed to be known at the receiver. The estimated CSI vector can $\hat{\mathbf{H}}_{MMSE} = \left[ \hat{h}_{1,m}, \ldots, \hat{h}_{n,m} \right]$ be estimated by

$$\hat{\mathbf{H}}_{MMSE} = \mathbf{R}_{HH} \left[ \mathbf{R}_{HH} + \left( \mathbf{SS}^H \right)^{-1} \sigma^2 \right]^{-1} \mathbf{S}^{-1} \mathbf{Y}$$ (4)

where $\mathbf{S} = [\hat{s}_{1,m}, \ldots, \hat{s}_{n,m}]$ and $\mathbf{Y} = [y_{1,m}, \ldots, y_{n,m}]$ are the vector of the transmitted and the received symbols, respectively. $\sigma^2$ is the power density of the noise. $\mathbf{R}_{HH}$ is the covariance of the channel frequency response (CFR) at the pilot tones.

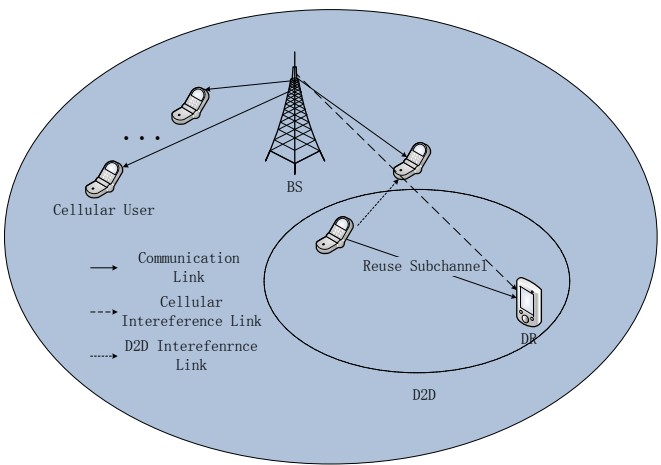

(**a**) Cellular layout and interference illustration at device-to-device (D2D).

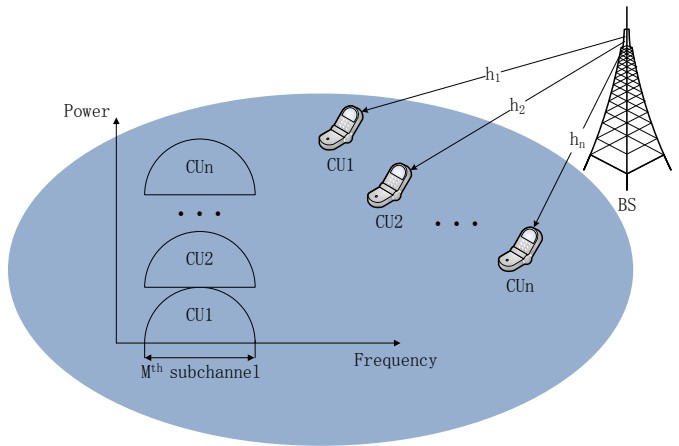

(**b**) Cell users (CUs) downlink non-orthogonal multiple access (NOMA) system.

**Figure 1.** Illustration of the NOMA-based cellular network with D2D. (**a**) Interference of cellular and D2D users. (**b**) Power versus frequency on each subchannel.

### 2.2. System Description

In the NOMA system, the BS transmits a multiuser signal to the CUs (as shown in Figure 1b), that comes from the same SC. When a CU receives the multiuser signal, the signal with the maximum power is first detected and eliminated by SIC the technology. The received signals are processed according to a descending sort of their power assigned by BS which is related to their own channel states.

As the channel gain of the users' increases, the power allocated to the CUs decreases. To give an illustration, the CUs at the edge of the cell usually have poor channel states and thus they are allocated more power. Therefore, the receiving signal of CU*i* can be divided into low power signals interference and high power signals interference of multiple access. So we note that (1) can be rewritten as

$$
\begin{aligned}
y_{i,m} = \quad & \hat{h}_{i,m} s_{i,m} \sqrt{p_{i,m}} + \hat{h}_{i,m} \sum_{k=1}^{i-1} s_{k,m} \sqrt{p_{k,m}} \\
& + \hat{h}_{i,m} \sum_{k=i+1}^{n} s_{k,m} \sqrt{p_{k,m}} + z_{i,m}
\end{aligned}
\tag{5}
$$

where $s_{k,m}$ is the transmitting signal for CU$k$, $p_{k,m}$ represents the transmit power for CU$k$. And $p_{k,m}$ follows an order of decreasing under a hypothesis that $\left|\hat{h}_{1,m}\right| \leq \left|\hat{h}_{2,m}\right| \leq \cdots \leq \left|\hat{h}_{n,m}\right|$. In NOMA scheme, resource blocks multiplexed are non-orthogonal in power domain. The interferences from other devices need to be removed. Successive interference cancellation (SIC) technology is introduced in the digital signal processing (DSP) based receiver for interference removal. Inside receivers, the SIC receiver needs to decode all data streams whose fractional power ratio is greater than the receiver's power, then subtracts the interference from the original symbols. The SIC works on a level by level manner, so the receiver should remove the interference from the data stream with the largest power, then remove the interference from the second largest, and so on.

After the SIC, the CUs with larger power are removed. Therefore, the superposition symbol can be simplified as

$$y_{i,m} = \hat{h}_{i,m}s_{i,m}\sqrt{p_{i,m}} + \hat{h}_{i,m}\sum_{k=i+1}^{n} s_{k,m}\sqrt{p_{k,m}} + z_{i,m} \tag{6}$$

where $h_{i,m}\sum_{k=i+1}^{n} s_{k,m}\sqrt{p_{k,m}}$ represents the interference from the part of the CUs with lower power on the same SC.

By using SIC, the received signal-to-interference-and-noise ratio (SINR) at CU$i$ can be written as

$$\text{SINR}_{i,m} = \frac{p_{i,m}\left|\hat{h}_{i,m}\right|^2}{\left|\hat{h}_{i,m}\right|^2\sum_{k=i+1}^{n} p_{k,m} + \sigma^2} = \frac{\beta_{i,m}}{\sum_{k=i+1}^{n} \beta_{k,m} + \frac{1}{\text{SNR}_{i,m}}} \tag{7}$$

where $\beta_{k,m}$ represents the $k-$th cell user's power allocation coefficient on SC$m$ and the signal-to-noise ratio (SNR) of the receiver CU$i$ is

$$\text{SNR}_{i,m} = \frac{\left|\hat{h}_{i,m}\right|^2 p_{BS,m}}{\sigma^2} \tag{8}$$

The total power from the BS to the CUs is denoted by $p_{BS,m}$ and we have $p_{k,m} = \beta_{k,m}p_{BS,m}$, $\forall k \in N$ which denotes the power allocated to CU$k$ on SC$m$.

From the Equation (7), we can obtain the achievable rate of CU$i$ of $m-$th SC

$$R_{i,m} = \log_2(1 + \text{SINR}_{i,m}) \tag{9}$$

From (7) and (9), we can observe that the SINR is determined by the power allocation coefficient$\beta$. By adjusting it, the BS can flexibly control the throughput or the achievable transmission rate of each user to optimize the performance of the system.

We assume $\left|\hat{h}_{1,m}\right| \leq \left|\hat{h}_{2,m}\right| \leq \cdots \leq \left|\hat{h}_{n,m}\right|$ and the power allocation coefficients become $\beta_{1,m} \geq \beta_{2,m} \geq \cdots \geq \beta_{n,m}$, then CU$i$ can decode and remove the interference from CU $j$, $\forall j < i$ successfully through SIC. However, the existing D2D devices also potentially contribute to the co-channel interference which affect the NOMA decoding order. To consider the interference from D2D, we can rewritten the received SINR at CU$i$ to decode the signal $s_{j,m}$, $j < i$, on SC$m$ as

$$\text{SINR}_{i \rightarrow j} = \frac{p_{j,m}\left|\hat{h}_{i,m}\right|^2}{\left|\hat{h}_{i,m}\right|^2\sum_{k=j+1}^{n} p_{k,m} + \sum_{l=1}^{n} \alpha_l q_{l,m}\left|\hat{h}_{l,i,m}\right|^2 + \sigma^2} \tag{10}$$

where the binary variable $\alpha_l$ represents whether or not CU $l$, $\forall l \in N$ is assigned to a D2D user in the same SC, $q_{l,m}$ ($0 \leq q_{l,m} \leq q_{l,\max}$) denotes the transmit power of the D2D pair with $q_{l,\max} = p_{BS,m}/n$, and $h_{l,i,m}$ is the channel gain from the $l-$th D2D pair to CU$i$ on SC$m$. And in the same way, we can get the received SINR at CU$j$ to decode its own signal $s_j$,

$$\text{SINR}_{j \rightarrow j} = \frac{p_{j,m}\left|\hat{h}_{j,m}\right|^2}{\left|\hat{h}_{j,m}\right|^2\sum_{k=j+1}^{n} p_{k,m} + \sum_{l=1}^{n} \alpha_l q_{l,m}\left|\hat{h}_{l,j,m}\right|^2 + \sigma^2} \tag{11}$$

where $h_{j,m}$ denotes the channel gain from BS to CU$j$ on SC$m$ and $h_{l,j,m}$ is the channel gain from the $l-$th D2D pair to CU$j$ on SC$m$.

Because admitting the access of multiple D2D devices brings a heavy signaling overhead into the system, we suppose that there is at most one D2D pair assigned to the same CU. The constraint of the power allocation coefficient is:

$$\sum_{l=1}^{n} \alpha_l \leq n, \alpha_l \in \{0, 1\} \tag{12}$$

When $\text{SINR}_{i \rightarrow j} \geq \text{SINR}_{j \rightarrow j}$, the received SINR from CU$i$ is no less than CU$j$'s received SINR and the interference can be successfully canceled by the SIC. According to the given SIC decoding order, the interference is always canceled from the CU with the largest power and the following conditions should be satisfied.

$$\frac{\sum_{l=1}^{n} \alpha_l q_{l,m} |\hat{h}_{l,j,m}|^2 + \sigma^2}{|\hat{h}_{j,m}|^2} \geq \frac{\sum_{l=1}^{n} \alpha_l q_{l,m} |\hat{h}_{l,i,m}|^2 + \sigma^2}{|\hat{h}_{i,m}|^2} \tag{13}$$

for $i, j \in \{1, \dots, n\} \triangleq N, j < i$, and the set $N$ represents the set of CUs' index on each SC. For $i \in N$, we note that the in Equation (11) can be simplified to

$$\frac{\sum_{l=1}^{n} \alpha_l q_{l,m} |\hat{h}_{l,i,m}|^2 + \sigma^2}{|\hat{h}_{i,m}|^2} \geq \frac{\sum_{l=1}^{n} \alpha_l q_{l,m} |\hat{h}_{l,i+1,m}|^2 + \sigma^2}{|\hat{h}_{i+1,m}|^2} \tag{14}$$

Also, we define the SINR of the $l-$th D2D device on the $m-$th downlink SC as

$$\text{SINR}_{l,m} = \frac{q_{l,m} |g_{l,m}|^2}{|\hat{h}_{BS,m}|^2 \sum_{k=1}^{n} p_{i,m} + \sigma^2} \tag{15}$$

where $h_{BS,m}$ is the interference channel gain from BS to D2D devices.

## 3. Proposed Joint User Scheduling, Tree-Based Search Power Allocation and Link Selection Algorithm

As discussed in the last section, we consider the SINR of the D2D devices along with the CUs and the system sum rate in the cellular network, in which the power to be allocated to the users is still unknown. To jointly solve the user scheduling, power allocation and link selection problems for the NOMA-based D2D enhanced cellular network, we propose the user scheduling, tree-based search and distance-aware link selection algorithm. In this section, we first discuss the impact of introducing D2D into the cellular network and give the system formulation. In addition, we present the power constraints of the D2D devices to access them into the network and give the data rates of the users. Secondly, we schedule users for improving the system throughput. Then, we propose the principles and steps of the tree-based search algorithm (TSA) to solve the power allocation problem of the entire network. Finally, we consider the subproblem of link selection to guarantee the communication quality between BS and D2D devices.

### 3.1. System Formulation

Firstly, we formulate the data rates of all the users in the network, including the CUs and the D2D users (DUs). Secondly, we discuss principles to allocate the SCs to the users to maximize the sum data rate of the system. Then we consider how to optimize the power allocation problem for the CUs.

Referring to (7) and (11), we can get a combination formula of the SINR of the NOMA-based D2D enhanced cellular network.

$$\text{SINR}_{i,m} = \frac{p_{i,m} |\hat{h}_{i,m}|^2}{|\hat{h}_{i,m}|^2 \sum_{k=i+1}^{n} p_{k,m} + \sum_{l=1}^{n} \alpha_l q_{l,m} |\hat{h}_{l,i,m}|^2 + \sigma^2} \tag{16}$$

where $p_{i,m}$ and $p_{k,m}$ represent the transmit power for CU$i$ and CU$k$, and $h_{i,m}$ denotes the channel gain from BS to CU$i$ on the $m-$th SC, $q_{l,m}$ denotes the transmit power of the D2D device, and $h_{l,i,m}$ is the channel gain from the D2D user to CU$i$ on SC$m$. $l$ is determined in $\alpha_l = 1$, such that the $l-$th D2D receiver is accessed to the CU$i$.

Note that $p_{k,m} = \beta_{k,m} p_{BS,m}$, $0 \leq \beta_{k,m} \leq 1$ and integrate (15) into (16), then we have

$$\text{SINR}_{i,m} = \frac{\beta_{i,m}}{\sum_{k=i+1}^{n} \beta_{k,m} + \frac{q_{l,m}|\hat{h}_{l,m}|^2}{p_{BS,m}|\hat{h}_{i,m}|^2} + \frac{1}{\text{SNR}_{i,m}}} \tag{17}$$

where the total power from the BS to the CUs on the same SC is denoted by $p_{BS,m}$. There are two variables $\beta_{i,m}$ and $q_{l,m}$ in Formula (17) which represent the power allocation coefficient of CU$i$ and the power of the $l-$th DU respectively on the $m-$th SC. As a hypothesis, if we fix the value of $\beta_{k,m}$, the SINR of CU$i$ is inversely proportional to $q_{l,m}$ when other parameters remain unchanged.

Then we consider the data rate of the DUs. To get the minimum transmission power, we need to obtain the constraint condition of $\text{SINR}_{l,m}^{D2D}$, i.e.,

$$\text{SINR}_{l,m}^{D2D} = \frac{q_{l,m}|\hat{h}_{l,m}|^2}{\sum_{k=1,k\neq l}^{n} p_{k,m}|\hat{h}_{k,l,m}|^2 + \sigma^2} \geq \text{SINR}_{thr}^{D2D} \tag{18}$$

where $\text{SINR}_{l,m}^{D2D}$ is the SINR of the $l-$th DU on the $m-$th SC, $\text{SINR}_{thr}^{D2D}$ denotes the given threshold for the DUs, $h_{l,m}$ denotes the channel gain DU$l$ on the $m-$th SC, $p_{k,m}$ represents the transmit power for CU $k$, and $q_{l,m}$ denotes the transmit power of DU$l$ on the $m-$th SC, $h_{k,l}$ denotes the interference channel gain of other CUs. Under the condition that $\text{SINR}_{l,m}^{D2D} = \text{SINR}_{thr}^{D2D}$, we can get the minimum transmission power of the DUs.

Based on the expression of SINR in (17) and the Shannon formula, the data rate for the $i-$th cellular user CU$i$ on the $m-$th SC is given by

$$R_{i,m} = \log_2\left(1 + \frac{\beta_{i,m}}{\sum_{k=i+1}^{n} \beta_{k,m} + \frac{q_{l,m}|\hat{h}_{l,m}|^2}{p_{BS,m}|\hat{h}_{i,m}|^2} + \frac{1}{\text{SNR}_{i,m}}}\right) \tag{19}$$

Similarly, the data transmission rate for the D2D device is written as

$$R_{l,m}^{D2D} = \log_2\left(1 + \frac{q_{l,m}|\hat{h}_{l,m}|^2}{\sum_{k=1,k\neq l}^{n} p_{k,m}|\hat{h}_{k,l,m}|^2 + \sigma^2}\right) \tag{20}$$

*3.2. User Scheduling Algorithm of the Network*

In this section, we design a user scheduling algorithm for assigning CUs and DUs to different SCs in order to maximize the system data rate. As shown in Algorithm 1, the algorithm also includes a power allocation problem, which is solved in the next section.

We denote $N = \{1, \ldots, i, \ldots, n\}$ as the set of CUs and define $L = \{1, \ldots, l, \ldots, L\}$ as the set of DUs. We assume that the maximum number of the D2D users $L$ is no more than $n$. To optimize the system data rate, we define the expression of the user scheduling as

$$U_{optimal} = \text{argmax}(R_C + R_D), \tag{21}$$

where $R_C = \sum_{m=1}^{M} \sum_{i=1}^{n_m} R_{i,m}$ and $R_D = \sum_{m=1}^{M} \sum_{l=1}^{L_m} R_{l,m}$ represent the sum rate of CUs and the sum rate of DUs respectively. $n_m$ and $L_m$ denote the number of CU and DU on SC$m$ respectively.

In Algorithm 1, we firstly schedule the CUs onto each SC. We assume equal power allocation for each CU and define $U_{max}$ as the maximum number of CUs allocated on each SC. $U_{un}$ is initialized to record the CUs who have not been allocated to any SC. In the scheduling procedure for CUs, we need to find the CUs who have larger channel gains than others and allocate them to the corresponding SCs if the number of CUs multiplexed on this SC is less than $U_{max}$. If the number of CUs is equal to $U_{max}$, the CUs should be selected from the CU sets $U_{m,possible}$, which includes all the CUs who prefer to be assigned on the $m-$th SC. The CUs who can provide the maximum data rates on this SC will be allocated on this SC. The CUs who have not been allocated on the SC will be returned to $U_{un}$. This part of algorithm ends when there is no CU left to be assigned [42].

---

**Algorithm 1** A User Scheduling Algorithm

---

1: Initialize the power allocation for each CU $P_{i,m} = \frac{P_{BS}}{n}$.
2: Construct the channel gains $\mathrm{H} \triangleq \left[\left|h_{i,m}\right|\right]_{n \times M}$.
3: Initial the sets $U_{un}$ to record the unallocated CUs in the system.
4: Initial the sets $U_{m,possible}$ to record the candidate CUs in the $m-$th SC.
5: Initial the sets $U_{un}^{D2D}$ to record the unallocated DUs in the system.
6: Initial the sets $U_{m,possible}^{D2D}$ to record the candidate DUs in the $m-$th SC.
7: **while** $U_{un} \neq \varnothing$ **do**
8:  Find the maximum value $\left|h_{i,m}\right|$ in H using $\left|h_{i,m}\right| = \arg \max\limits_{i \in U_{un}, m \in H_{un}} (\mathrm{H})$.
9: **if** the number of multiplexed CUs on this SC is less than $U_{max}$ **then**
10: (a) Schedule the CU$i$ onto the SC$m$.
11: (b) $U_{un} = U_{un} \backslash U_i$.
12: **end if**
13: **if** the number of multiplexed CUs on this SC equals $U_{max}$ **then**
14: (a) Assume CU$i$ is allocated on the SC$m$ and the CU set is $U_{m,possible}$.
15: (b) Calculate the data rate of the CUs from $U_{m,possible}$ and get a set of $R_{m,possible}$.
16: (c) $U_m = \arg \max\limits_{U \in U_{m,possible}} \left(R_{m,possible}\right)$ and $U_i \notin U_m$.
17: (d) $U_{un} = U_{un} \backslash U_m$.
18: Let the $i-$th and $n-$th row's elements in H be zeros.
19: Let the $n-$th column's elements in H be zeros.
20: **end if**
21: **end while**
22: Allocate power to the CUs (Algorithm 2).
23: Construct the channel gains $\mathrm{H}^{D2D} \triangleq \left[\left|h_{l,m}\right|\right]_{L \times M}$.
24: Initialize the power allocation for each DU.
25: **while** $U_{un}^{D2D} \neq \varnothing$ **do**
26: Find the maximum value $\left|h_{l,m}\right|$ in $\mathrm{H}^{D2D}$ using $\left|h_{l,m}\right| = \arg \max\limits_{l \in U_{un}^{D2D}, m \in \mathrm{H}_{un}^{D2D}} (\mathrm{H}^{D2D})$.
27: **if** the number of multiplexed DUs on this SC is less than $U_{max}^{D2D}$ **then**
28: (a) Schedule the DU$l$ onto the SC$m$.
29: (b) $U_{un}^{D2D} = U_{un}^{D2D} \backslash U_l$
30: **end if**
31: **if** the number of multiplexed DUs on this SC equals $U_{max}^{D2D}$ **then**
32: (a) Assume DU$l$ is allocated on the SC$m$ and the DU set is $U_{m,possible}^{D2D}$.
33: (b) Calculate the data rate of the DUs from $U_{m,possible}^{D2D}$ and get a set of $R_{m,possible}^{D2D}$.
34: (c) $U_m^{D2D} = \arg \max\limits_{U \in U_{m,possible}^{D2D}} \left(R_{m,possible}^{D2D}\right)$ and $U_i^{D2D} \notin U_m^{D2D}$.
35: (d) $U_{un}^{D2D} = U_{un}^{D2D} \backslash U_m^{D2D}$.
36: Let the $i-$th and $n-$th row's elements in $\mathrm{H}^{D2D}$ be zeros.
37: Let the $n-$th column's elements in $\mathrm{H}^{D2D}$ be zeros.
38: **end if**
39: **end while**

---

Then we allocate power to the CUs of each SC, which can be updated by Algorithm 2.

Finally, we assigned the DUs of the system into each SC. In this part, similar to CUs, we assume that each DU has the same power and define $U_{\max}^{D2D}$ $\left(U_{\max}^{D2D} \leq U_{\max}\right)$ as the maximum number of DUs allocated on each SC. $U_{un}^{D2D}$ is initialized to record the DUs who have not been allocated to any SC. We calculate the transmission power of each DU by (18) with the condition that $\text{SINR}_{l,m}^{D2D} = \text{SINR}_{thr}^{D2D}$. In the scheduling process, we need to find the DUs which have larger channel gains than others and access them to the corresponding CUs if the number of CUs multiplexed on this SC is less than $U_{\max}^{D2D}$. If the number of DUs is equal to $U_{\max}^{D2D}$, the DUs should be selected from the DU sets $U_{m,possible}^{D2D}$, which includes all the DUs who prefer to be assigned on the $m-$th SC. The DUs which can provide the maximum data rates in this SC will be allocated on this SC. The DUs which have not been allocated on the SC will be returned to $U_{un}^{D2D}$. This part of algorithm ends when there is no DU left to be allocated.

Since the maximum CUs and DUs can be multiplexed on the same SC is less than $U_{\max}$, the global optimal solution can only be obtained by the exhaustive search method (ESA), which has exponential complexity with respect to the number of SCs. In order to lower the computational complexity and increase the processing speed, we propose a tree-based power allocation scheme to for accelerating the searching progress in the next part. The convergence of the proposed algorithm is shown in Section 4.

*3.3. Principles of Tree-Based Search Power Allocation Algorithm*

In this section, we suppose that the power $q_l$ assigned to the transmission from the transmitter CU$l$ to receiver (DR) in each D2D group is a fixed value which can be calculated by (20) with the condition that $\text{SINR}_{l,m}^{D2D} = \text{SINR}_{thr}^{D2D}$.

Note that, by NOMA technology, some access algorithms can allocate power in a gradient increasing way with the deterioration of the users' channel state. This method simply calculates the representation of the objective function and need not derivative in the process of solving the problem. In order to reduce the cost of computing compared with [30] and [32], we propose a tree-based search power allocation algorithm, which gives a suboptimal method to maximize the system sum rate.

In NOMA, the main interference is from low power signals which cannot be eliminated by SIC. Through the decoding order, because the channel gain of a CU in the cell center is larger, a lower power is allocated to the CU. Thus, by using the SIC technology, other users can gain greater SINR after eliminating the interference of the CUs with higher power and hence have better system performance.

We set the target of power allocation to be maximizing the arithmetic average of achievable user data transmission rates of each SC. The optimization problem can be obtained as

$$\left\{\beta_{1,m}, \beta_{2,m}, \cdots, \beta_{n_m,m}\right\} = \text{argmax}\left\{\frac{1}{n_m}\sum\nolimits_{k=1}^{n_m} R_{k,m}\right\}, \tag{22}$$

$$s.t.\ 0 < \beta_{i,m} < 1, i = 1, 2, \cdots, n_m \tag{23}$$

$$\beta_{i,m} \leq \beta_{j,m}, i = 1, 2, \cdots, n_m - 1, i < j \tag{24}$$

$$\sum\nolimits_{i=1}^{n_m} \beta_{i,m} = 1 \tag{25}$$

where $\beta_{i,m}$ represents a proportional coefficient of the power value that can be allocated to each CU on SC$m$. (22) is the objective function and constraint (23) is imposed to guarantee the power allocated to the CUs will not exceed the total power from the BS. Constraint (24) shows the power allocated to the CUs of the same SC follow the policy for SIC decoding order. Constraint (25) presents that all the power can be allocated to the CUs. $n_m$ is the max number of CUs on SC$m$.

In the tree model of TSA, we need to explain several concepts. The first one is the node of the tree which represents a candidate coefficient β of the assigned power. The second one is the depth of a node which is defined by the length of the path from the root to the node. In addition, the nodes with the same depth are identified to be in the same layer and the quantity of the layers depends on the number of the non-orthogonal users on the same SC. Then we define the branches, each of which connects the

nodes of two adjacent layers. The tree model of TSA is shown in Figure 2. $\beta_{k,m}, \forall k \in N$ represents the power coefficients allocated to the users in the $k-$th layer on SC$m$. In a descendant order of the SINR of the CUs according to (5), the coefficients $\beta$ also satisfy (23)–(25), which means the coefficient in current layer cannot be higher than the next layer and cannot be lower than the previous layer at the same time. Two specific conditions that need to be met are shown as follows:

$$0 < \beta_{1,m} \leq \frac{1}{n_m} \tag{26}$$

$$\beta_{k-1,m} \leq \beta_{k,m} \leq \frac{1 - \sum_{j=1}^{k-1} \beta_{j,m}}{n_m - k + 1} \tag{27}$$

where $k \in N$.

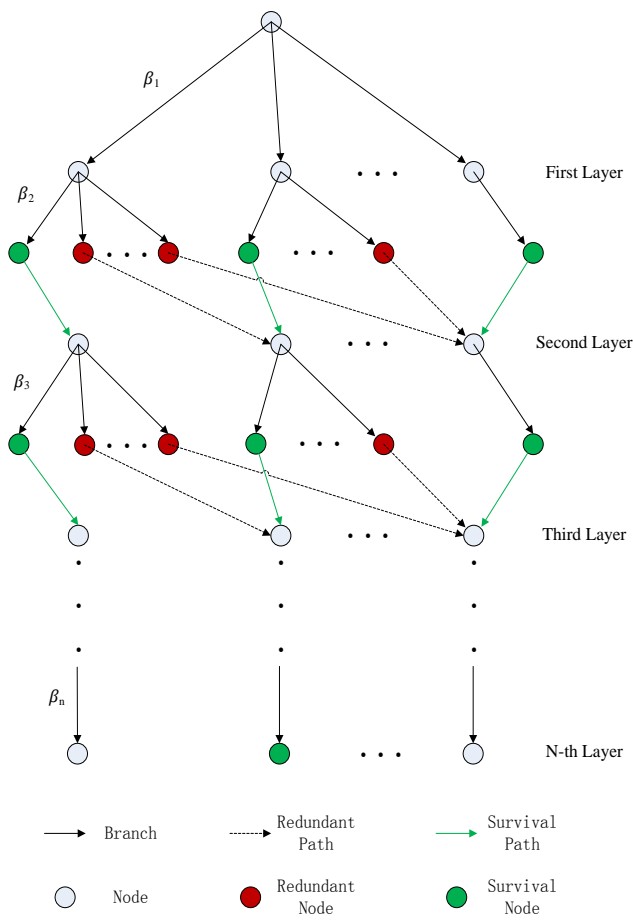

**Figure 2.** Propose a tree model and find a final path to allocate power to each CU through the tree-based search method. In each layer the nodes are divided into several groups to pick out the best node for the next layer.

In Figure 2, the power allocated to the CUs in each layer is in the order from low to high. The survival nodes and the redundant nodes in the tree represent whether the power assigned to the current CU can satisfy the requirement of the target function. The path through the redundant node is called a redundant path which does not undergo the search process of the next layer. Therefore, it reduces the searching complexity compared with the ESA. Similarly, the paths through the survival nodes are survival paths. Finally, the longest path becomes the final path containing all the power allocation coefficients of the CUs every layer.

As introduced in (3), after using the SIC technology, the target CU is influenced only by the interference coming from the CUs with low power. Different from traditional tree-based search algorithms [43–46], we arrange the CUs in a descending order of SINR in a tree and allocate power to the CUs in an increasing order. Thereby, we draw the conclusion that the CUs would only be influenced by the signals from CUs in lower layers rather than in higher layers, which can reduce the number of searches.

*3.4. Proposed Tree-Based Search Algorithm (TSA) for Power Allocation*

In this section, we first define two kinds of judgement standards for selecting the candidate nodes and decide which nodes should result in survival. One of the standards about the power allocation coefficients $\beta_{k,m}$ is

$$\Omega_{s,m} = \sum_{k=1}^{s} \beta_{k,m}, \ s \in n_m \tag{28}$$

which represents the summary of $\beta$ from the first layer to the $s-$th layer. For $n_m \in N$, the number of the CUs on each SC, we have $\Omega_{n_m} = 1$ according to (25). Another standard is for the data rates,

$$\Gamma_{s,m} = \sum_{k=1}^{s} R_{s,m}, \ s \in n_m \tag{29}$$

which denotes the summary of the data rates of CUs from 1 to $s$. For $n_m$, $\Gamma_{n_m}$ represents the summary of the data rates from CU 1 to CU$n_m$.

The TSA achieves the power allocation of the CUs within each SC and finds a suboptimal solution, as shown in Algorithm 2. The detailed process of TSA is described as the following steps:

(1) Initialization: First, we initialize the standards (28) and (29) mentioned above and the channel gain of each CU on the same SC.
(2) Calculate the values of the power allocation coefficients: Calculate the matrix of candidate power allocation coefficients $\beta_{k,m}$ in the $k-$th layer, according to (26) and (27), determined by the number of survival nodes of the previous layer and the minimum interval $\Delta$ of the power allocation coefficients.
(3) Delete the redundant nodes: We first divide the nodes into several groups. In each group, through (26), we calculate $\Omega_{k,m} = \Omega_{k-1,m} + \beta_{k,m}$ and find out the nodes belong to the $k-$th layer which has the same $\Omega_{k,m}$ to be classified to the same group. Then through the formulation $\Gamma_{k,m} = \Gamma_{k-1,m} + R_{k,m}$ derived from (26), we get the results of $\Gamma_{k,m}$ in each group and pick out the maximum nodes. Finally, we select the survival nodes and delete the redundant paths.
(4) Select the final survival path: Repeat step (2) and step (3) until the last layer where $\beta_{n_m} = 1 - \Omega_{n_m-1}$ and the number of the branches equals to the number of the survival nodes in the previous layer. Satisfying the unique value of $\Omega_{n_m}$, i.e., $\Omega_{n_m} = 1$, we pick out the group of survival nodes with the maximum $\Gamma_{n_m}$ and get the final survival path.
(5) Output: From final survival path we can get the suboptimal set of power allocation coefficients $\{\beta_1^*, \cdots, \beta_{n_m}^*\}$.

Figure 3 shows the flow chart of the tree-based search power allocation algorithm (TSA). For the suboptimal solution from Algorithm 1, the combinations of the searching process would be

$$M! \left\{ \left( \binom{n}{1} + \binom{n}{2} + \cdots + \binom{n}{U_{\max}} \right) + \left( \binom{L}{1} + \binom{L}{2} + \cdots + \binom{L}{U_{\max}^{D2D}} \right) \right\}, \tag{30}$$

On the other hand, the complexity of the proposed algorithm is less than the complexity of ESA which would be

$$O\left(N! \left(2^n + 2^L\right)\right), \tag{31}$$

Figure 4 investigates the comparison of the computational complexity cost between TSA and ESA with $\Delta = 0.01$ which represents the minimum interval of the power allocation coefficients [38]. Because NOMA allows the receiver to remove the part of the interference coming from the users who have larger power, it decreases the search times for each user. The complexity cost of TSA is $O\left(\frac{1}{\Delta^2}\left(1 - \frac{1}{n}\right)\right)$ and this is lower than ESA whose complexity cost is $O\left(\frac{1}{\Delta^{n-1}}\right)$ [31]. The total computational complexity cost of the proposed TSA algorithm with the searching process is

$$O\left(\frac{1}{\Delta^2}\left(1 - \frac{1}{n}\right)\right) + M!\left\{\left(\begin{pmatrix} n \\ 1 \end{pmatrix} + \cdots + \begin{pmatrix} n \\ U_{\max} \end{pmatrix}\right) + \left(\begin{pmatrix} L \\ 1 \end{pmatrix} + \cdots + \begin{pmatrix} L \\ U_{\max}^{D2D} \end{pmatrix}\right)\right\},$$

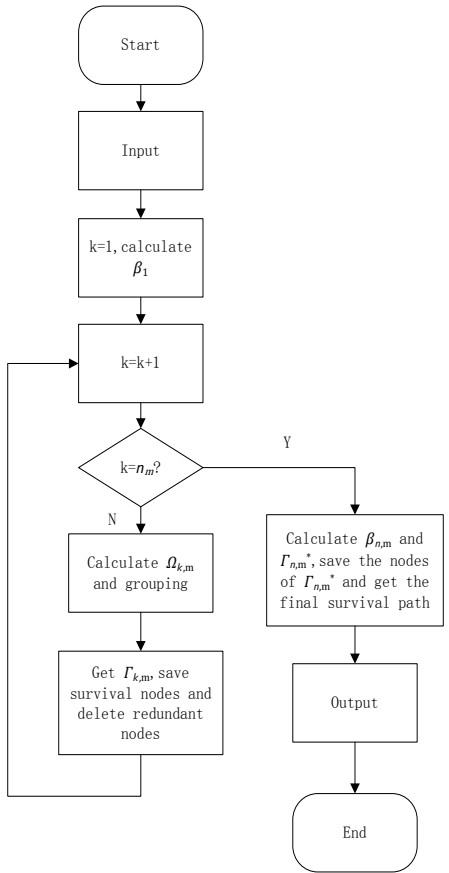

**Figure 3.** Flow chart of the tree-based search power allocation algorithm (TSA).

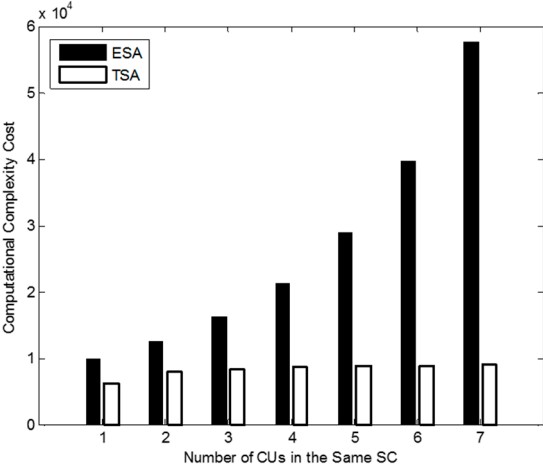

**Figure 4.** Computational complexity cost comparison of the tree-based search algorithm and the exhaustive search algorithm with $\Delta = 0.01$.

---

**Algorithm 2** Tree-based Search Power Allocation Algorithm (TSA)

---

1: *Input:*
- Initial the judgement criteria: $\Omega_0 = 0$ and $\Gamma_0 = 0$.
- Initial list of the channel states of the CUs in the same SC $\{h_{1,m}, \ldots, h_{n,m}\}$.

2: *Tree-based search:*

　**for** $k \in \{1, \ldots, n_m\}$ **do**

　**repeat**

　Calculate $\beta_{k,m}$ according to survival nodes of the previous layer and the minimum interval $\Delta$ of the power allocation coefficients which satisfies the condition (26) and (27).

　Calculate $\Omega_k$ and find out the nodes of the same $\Omega_k$ to put them into $z$ groups.

　　**for** $r \in \{1, \ldots, z\}$ **do**

　　　Calculate the set of $\Gamma_r$ and $\Gamma_r^* = \max \{ \}$ and make the new set of $\Gamma_k$. Save the nodes of $\Gamma_r^*$ and delete the others.

　　**end for**

　**until** $k = n_m$.

　**If** $k = n_m$ **then**

　　Calculate $\beta_{n_m}$ and $g = 1$. $\Gamma_{n,m}^* = \max \{\Gamma_{n,m}\}$. Save the nodes of $\Gamma_{n,m}^*$ and delete the others. Finally get the final survival path.

　**end if**

　**end for**

3: *Output:*

　Final set of power allocation coefficients $\left\{\beta_1^*, \ldots, \beta_{n_m}^*\right\}$.

---

### 3.5. Distance-Aware Link Selection Algorithm

In this section, we design a link selection algorithm for accessing DUs to different CUs on different SCs in order to further optimize the system data rate, as shown in Figure 5. The algorithm also influences the power allocation for DUs.

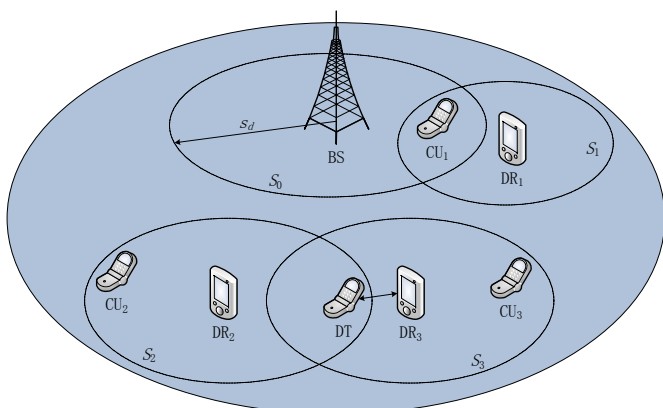

**Figure 5.** Model of the distance-aware link selection algorithm. In the figure, all the devices are serviced in the same cell by the same base station (BS).

In our model, a DU always chooses a CU to be the signal transmitter (DT) of the D2D pair, as mentioned in Section 2. Thus, the spectrum resources are reused by DUs and CUs of the same D2D pairs on the same SC. As the number of DU increases, the number of the spectrum reused links becomes large, leading to a high computational complexity cost. Different from the user scheduling scheme in Section 3, we select the D2D links by a distance-aware link selection algorithm to relieve complexity stress.

There are two principles in the algorithm. The first one is the distance between the DT and the BS cannot be too short. Based on the interference-limited area based scheme [47], BS has a limit circle

where the users cannot choose D2D links with radius $s_d$, as shown by $S_0$ in Figure 5. Because the DT may impose severe interference on BS and close-by CUs in the limit circle $S_0$, this principle can improve the QoS of users in the network.

The other principle is about the distance between DUs. The purpose of the principle, on the other hand, is to limit the co-spectrum interference. In this case, if two D2D receivers (DRs) are close enough that they prepare to choose the same CU to be the DT, the link which can provide better data rate would be chosen. The other DR can choose a relatively long distance link on different SCs.

The principles for choosing appropriate D2D links is illustrated in Figure 5 as follows:

(1)   $CU_1$ cannot be chosen as a DT by $DR_1$, because it is inside the limited area $S_0$.
(2)   In area $S_2$, $DR_2$ has two CUs to be selected to build a D2D link. But one of the CUs has already been chosen by $DR_3$ because the distance between the D2D pair is shorter. Thus, $DR_2$ can choose $CU_2$ to be the DT, in the same area $S_2$.
(3)   When two areas have a communal area (e.g., $S_2$ and $S_3$), the DRs may choose the same CU as the DT. In this case, we allocate them in different SCs to limit the co-spectrum interference.

### 3.6. Joint User Scheduling, Power Allocation and Link Selection Algorithm

Because the ESA, as shown in Figure 6a, has a high complexity cost in optimizing the user scheduling and power allocation problem, it is necessary to update the power of each user after changing the assignment of SCs in each iteration process. We propose the joint user scheduling, power allocation and link selection algorithm (JUPLA), as shown in Figure 6b, to be an alternative low-complexity approach. The proposed algorithm first solves the SC assignment problem, then allocates power to each CU and finally selects the links for DUs and outputs the network data rate. Figure 6 shows the flow chart of the proposed JUPLA.

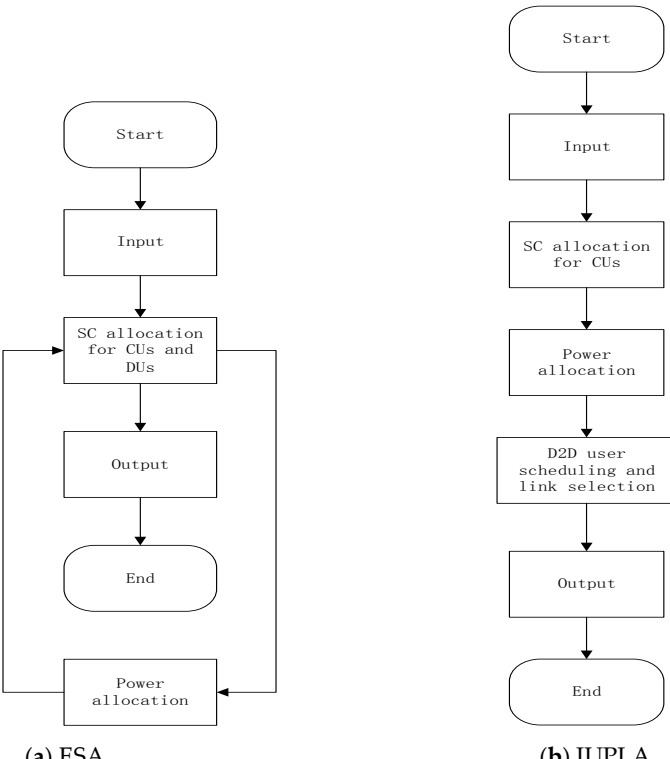

(**a**) ESA.                    (**b**) JUPLA.

**Figure 6.** Flow chart of the exhaustive search algorithm and the joint user scheduling, tree-based search power allocation and distance-aware link selection algorithm. (**a**) Flow chart of the exhaustive search algorithm (ESA). (**b**) Flow chart of the joint user scheduling, power allocation and link selection algorithm (JUPLA).

## 4. Numerical Results and Discussions

In this section, we present the performance of the proposed JSPLA through simulations. To evaluate the performances, a Monte Carlo based system-level simulator has been built. Each point of the simulation result is averaged over 1000 times. We first study the convergence performance of the proposed algorithm. Then we investigate the data rates of three search algorithms including the fractional power allocation algorithm (FRPA), the fixed power allocation algorithm (FIPA) and the TSA. Moreover, the performance comparison of the conventional ESA and OMA based D2D communications demonstrates the potential benefits of the proposed NOMA enhanced D2D scheme. The specific parameter value settings are summarized in Table 1.

**Table 1.** Propose the simulation parameters of the study.

| Parameter | Value |
| --- | --- |
| Cellular radius | 450 m |
| Maximum distance between D2D pairs | 75 m |
| Total bandwidth | 2.0 MHz |
| Carrier frequency | 2.6 GHz |
| Maximum Transmission power of CU | 21 dBm |
| Maximum transmission power of D2D | 21 dBm |
| Maximum transmission power of BS | 26 dBm |
| Number of CU | 10 |
| Number of D2D pairs | 10 |
| Number of users on each subchannel | 3 |
| SINR threshold of D2D | 1.8 dB |
| SINR threshold of CU | 1.3 dB |
| Noise Spectral density | −173 dBm |
| Path loss constant | 0.01 |
| Path-loss exponent | 3.76 |

### 4.1. Convergence of the Proposed Algorithm

The cumulative distribution function (CDF) versus the number of searching in JUPLA is shown in Figure 7. The results in Figure 7 are averaged over 10,000 independent adaptation processes which involve different numbers of users from 20 to 80 on 10 SCs. It can be observed that with $\Delta = 0.15$ the proposed joint resource allocation algorithm has a fast converge rate in 1000 iterations for all considered numbers of users. As the number of users decreases, the convergence rate slows down.

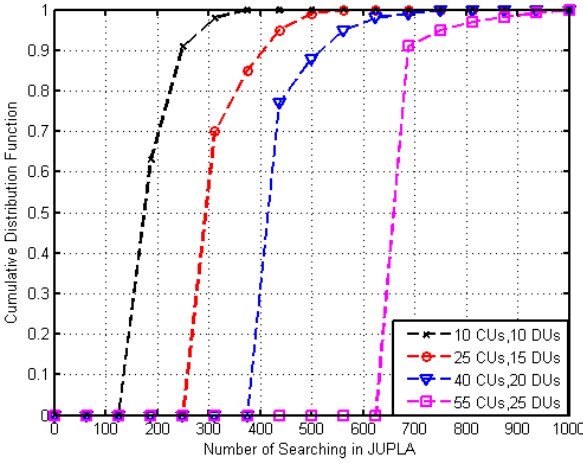

**Figure 7.** CDF of the number of branches in each search process, with $\Delta = 0.15$, which depicts the probability of finding out the solution increases with the growing of the search number.

### 4.2. NOMA-Enhanced Versus OMA-Based D2D Communication

In Figure 8, the average D2D data rate performance of three algorithms versus the number of the CUs on the same SC is shown in two accessing ways (NOMA and OMA). From Figure 8, the average data rate of the D2D devices decreases with the rising of the CU number on the same SC. Because of the increasing number of CUs, the probability of the D2D users receiving more interference from the CUs increases, which leads to the monotonous decreasing of data rates with the increasing of *n* according to (20). In addition, the NOMA based on D2D scheme achieves larger data rate than the conventional OMA based on D2D scheme. In NOMA, the performance of FIPA is worst, while that of FRPA is improved. Our proposed algorithm has the best data rate.

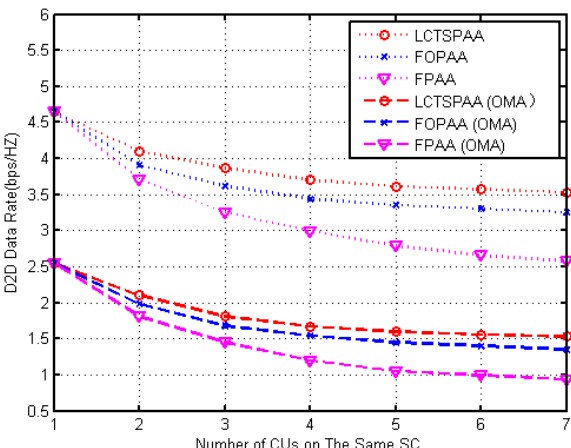

**Figure 8.** D2D data rate versus number of CUs on the same SC. We study how the number of users influences the data rates on each SC where the number of the users changes from 1 to 7.

### 4.3. Data Transmission Rates of CUs in the Same SC in the Network

Figure 9 illustrates that the increasing of the number of CUs on the same SC affects the data rates with arithmetic average. The arithmetic average can reflect the real average data rate of all the CUs in the network. In the four algorithms, TSA, FRPA, FIPA and fixed SINR power allocation algorithm (FSPA) [38], the arithmetic average of data rates of CUs decreases with a changing speed that gradually slows down as the number of the CUs increases.

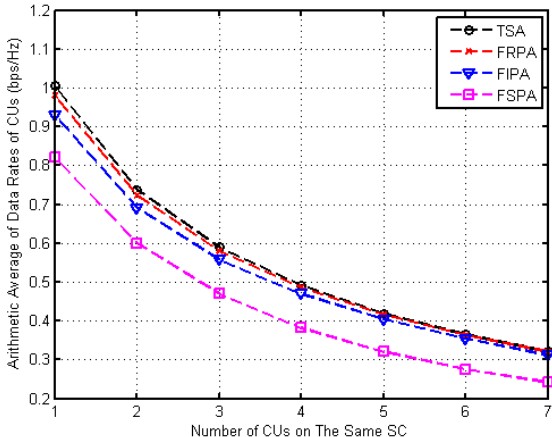

**Figure 9.** Arithmetic average of data rates versus number of CUs on the same SC. We study how the increasing of users influences the arithmetic average of the data rates on each SC where the number of the users changes from 1 to 7.

Figure 10 illustrates how the increasing of the number of CUs on the same SC affects the data rates with geometry average. In the four algorithms, the geometry average of data rates of CUs decreases with a changing speed that gradually slows down as the number of the CUs increases. For the geometric average, the standard of the data rates is

$$\Gamma_{s,m} = \prod_{k=1}^{s} R_{s,m} \tag{32}$$

which denotes the data rates of CUs from 1 to $s$. Noticing that the geometry average is less affected by mutation data than the arithmetic average, from two different points of view, we can comprehensively evaluate the simulation results.

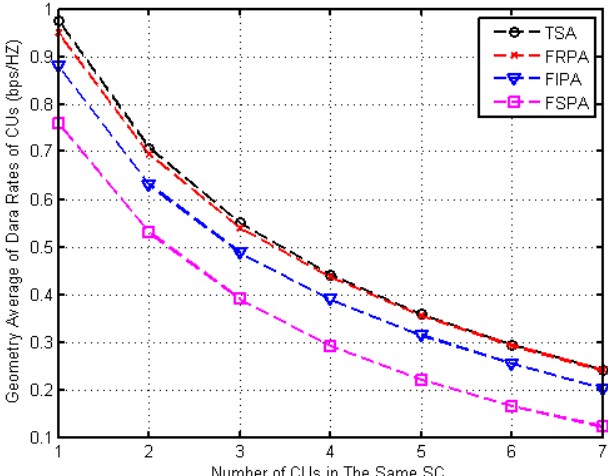

**Figure 10.** Geometry average of data rates versus number of CUs on the same SC. We study how the increasing of users impacts on the geometry average of data rates of CUs on each SC where the number of the users changes from 1 to 7.

From the two figures, we can note that the output performance is influenced by the number of users. The two figures also give comparison of the four algorithms to demonstrate the superiority of TSA. In FIPA and FSPA, the power is allocated to the CUs in fixed methods without considering the current channel states, whereas in FRPA the power is simply allocated referring to the path loss of each CU without concerning the sum rate of the whole system. In summary, the performance of FSPA is worst and FIPA and FRPA can improve the output performance. However, the proposed algorithm is the best of all the algorithms we considered.

*4.4. Sum Data Rate of CUs in the Same SC in the Network*

Figure 11 displays the sum data rate of the four algorithms versus the number of CUs on the same SC. In the four algorithms, the sum data rate of CUs increases with a changing speed that gradually slows down as the number of the CUs increases on the same SC. From Figure 11, the increment of sum data rate FSPA is smaller than the other three algorithms and the sum rate performance of FRPA becomes closer to TSA with the increasing of the number of CUs of the network.

Summarizing the results obtained from Figures 9–11, we observe that the increasing speed of the data rate slows down with the increase of the number of the non-orthogonal CUs. This is because, with the increasing of the number of CUs, more multiple access interference is introduced into the network and the SINR of each CU decreases along with the data rate.

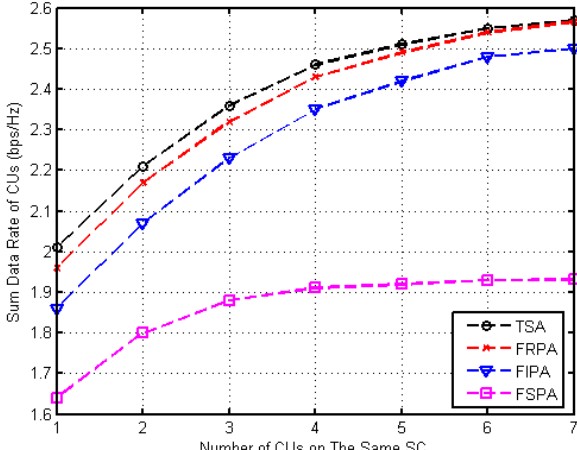

**Figure 11.** Plot of the sum data rate of CUs on the same SC with respect to the number of CUs. On each SC, the number of CUs changes from 1 to 7.

### 4.5. Sum Data Rate of the System

Figure 12 displays the sum data rate of the four algorithms versus the number of users in the network. In the four algorithms, the sum data rate of the system increases with a changing speed that gradually slows down as the number of users increases. When the user number increases and the total energy of the BS remains unchanged, the power of CUs is reduced. With the increasing number of D2D users, more CUs need to consume power on extra data transmission, which leads to further energy consumption. As the user number increases, the BS provides more SCs to the network. For example, when the user number is 80, there are 4 SCs in the network with 20 users in each SC. From Figure 12, the increment of sum data rate FSPA is smaller than the other three algorithms and the sum rate performance of FRPA becomes closer to TSA as the number of users of the system increases.

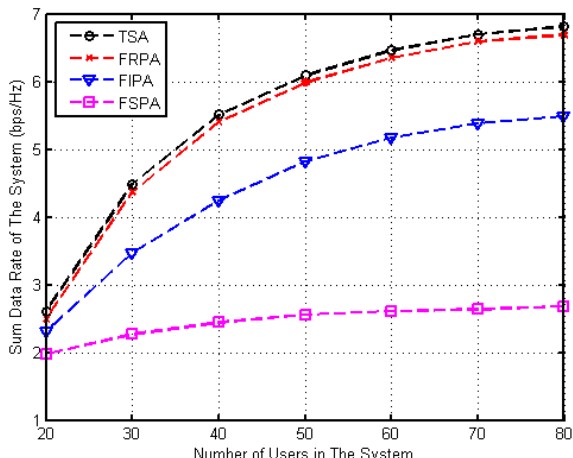

**Figure 12.** Plot of the sum data rate of the system versus the number of users.

## 5. Conclusions

In this paper, we proposed a joint user scheduling, tree-based search power allocation and link selection algorithm in building a NOMA-based D2D enhanced cellular network. Firstly, we introduced the D2D communication into the cellular network to increase the system capacity by reusing the spectrum of CUs. Secondly, the proposed algorithm jointly solved the user scheduling, power allocation and link selection problems to increase the sum data rate of the network. The user scheduling method and the tree-based search power allocation scheme were proposed not only to achieve the maximum data rate of the CUs and DUs, but also reduce the cost of computing compared with ESA. The distance-aware

link selection algorithm limits the interference for the DUs, saves energy for the CUs and offloads part of the heavy traffic for the BS. Finally, we conducted simulations to evaluate the performance of our proposed algorithm. Numerical results demonstrate that our algorithm improves data rate, has a low computational complexity cost and has a good convergence performance. Thus, the algorithm also has significant advantages compared with traditional algorithms.

**Author Contributions:** Conceptualization, J.W. and X.S.; methodology, J.W.; software, Y.M.; validation, J.W., X.S. and Y.M.; formal analysis, J.W.; investigation, X.S.; resources, Y.M.; data curation, X.S.; writing—original draft preparation, J.W.; writing—review and editing, J.W. and X.S.; visualization, J.W.; supervision, X.S.; project administration, X.S.; funding acquisition, X.S. All authors have read and agreed to the published version of the manuscript.

**Funding:** This research was funded by "the National Nature Science Foundation of China, grant number no. 61601109" and "The APC was funded by the Fundamental Research Funds for the Central Universities under Grant No. N152305001".

**Acknowledgments:** This work was supported by the National Nature Science Foundation of China under Grant no. 61473066 and no. 61601109, and the Fundamental Research Funds for the Central Universities under Grant No. N152305001. The authors thank the anonymous reviewers for their insightful comments that helped improve the quality of this study.

**Conflicts of Interest:** The authors declare no conflict of interest.

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
