# Peer review of "A Novel Resource Allocation Scheme in NOMA-Based Cellular Network with D2D Communications"

_futureinternet, doi:10.3390/fi12010008_

Round 1

Reviewer 1 Report

The reviewer's main concerns are as follows.

1. The assumption of perfect channel state information at the BS in the channel model is strong. Applying the minimum mean-square error (MMSE) channel estimation is suggested in the channel model.

2. How the SIC technology is modeled on the proposed solution?

3. The computation of TSA complexity cost is not clear enough. Is there any alternative metric for checking the efficiency of the proposed method? (such as the number of floating-point operations or asymptotic complexity analysis)?

4. In Section 4 and 5, authors should provide additional details about the simulator adopted to achieve performance results. In this manner, it would be possible to better assess the soundness of the proposed solution.

5. Some parts of the text are difficult to follow. The readability could be improved by adding more detailed explanations on the network model and system formulation.

6. The quality of the figures must be improved. Proofreading is required to fix some grammar and typo errors.

Author Response

Dear Editor,

RE: futureinternet-649148

We would like to thank Future Internet for giving us the opportunity to revise our manuscript.

Best wishes,

Mr. Wang, Prof. Song and Mr. Ma

Reviewer 2 Report

This paper presents a power allocation and link selection algorithm for 5G Device-to-Device communications with Non-Orthogonal Multiple Access (NOMA). The algorithm is described correctly and its performance is evaluated in simulations against similar alternatives. The work is solid, but there are some choices regarding the channel model that require clarification:

A channel model based on path loss has been used. What about multipath fading? In Table I a path loss exponent of 3.5 has been selected. Where does this value come from? It seems that the same type of wireless channel has been applied to both cellular (base station to device) and D2D (device to device) links. Does it make sense? Should they not be different types of channels?

Finally, some minor editing remarks:

In Section I: "the systems are combined" instead of "the systems are combine". In Section II.A: "a NOMA-based" instead of "an NOMA-based". In Section IV.C: "in fixed methods" instead of "in fix methods".
